# Factors related to the intention of choosing shared E-scooters for metro transfer: A survey study integrating weather perception into satisfaction evaluation from Changsha

**Chenyue Lin[1,2‡], Xingjian Xue**📧[1‡]*, **Zhixuan Zhu[1], Yue Luo[1,2], Rui Song**📧[2]

**1** Hunan Big Data Engineering Technology Research Center of Natural Protected Areas Landscape Resources, Central South University of Forestry and Technology, Changsha, Hunan, China, **2** College of Logistics and Transportation, Central South University of Forestry and Technology, Changsha, Hunan, China

‡ CL and XX contribute equally to this paper and should be considered as co-first author
* 7413442@qq.com

**Data Availability Statement:** All relevant data are within the paper and its Supporting Information files.

## Abstract

Shared E-scooter (SE) travel is a low-carbon transportation method that can be further enhanced by integrating with metro systems. This study aims to quantify the impact of the built environment, attitude preferences, weather perception, and other factors on the evaluation and intention to use the "SE-metro transfer" travel mode, as well as how to efficiently and concisely measure and model these effects. Empirical analysis was conducted using questionnaire data from Changsha, China, with 683 participants surveyed. Three satisfaction models were established and compared based on the Technology Acceptance Model (TAM), and an optimal M2 model was expanded to incorporate users' subjective perceptions of weather, proposing a method to simplify questionnaire length. The study found that well-designed vehicles and infrastructure, along with necessary supporting facilities, play important roles in enhancing SE usage. However, there are still many areas for optimization in Changsha's SE policies. Despite the advantages of SE in terrain and physical fitness, which have significantly expanded and changed their user base compared to traditional shared bicycles, there is still much potential to adapt to the middle-aged and older user groups. The results of this study can provide valuable insights for professionals and government officials in designing systems, constructing infrastructure, and formulating policies.

## 1 Introduction

In the field of transportation, which accounts for a quarter of global greenhouse gas emissions [1], China has formulated the "Dual Carbon" strategy to encourage the use of low-carbon transportation modes. Metro transit is a low-carbon transportation mode with large capacity, high speed, and exclusive right of way. However, the lack of station density limits people's choice of metro transit for travel. With the rise of shared bicycles, the service radius of metro stations has been effectively expanded [2], making it a crucial issue to promote the integration

**Funding:** Xingjian Xue received supports by the Natural Science Foundation of Hunan Province, China (Grant No. 2022JJ31017) . The funders had no role in study design, data collection and analysis, decision to publish, or preparation of the manuscript.

**Competing interests:** The authors have declared that no competing interests exist.

of shared bicycles with metro transit effectively. In mountainous and hilly cities in southern China, such as Changsha in this case study, shared e-scooters (SE) have largely replaced shared bicycles due to their advantages in speed, physical effort, hill climbing, and travel distance [3]. Research has shown that more than 40% of respondents are willing to adopt the "SE + metro transit" as their daily travel mode [4], indicating the enormous potential of this travel mode.

However, people's evaluations of this mode of transportation vary, various factors may cause this disparity. Existing literature has largely clarified the common factors influencing bike-sharing transfers to metro transit, including urban topography, weather and environment, infrastructure, service quality, demographic and socioeconomic characteristics, cognition and experience, as well as social norms (see Section 2.1 for details). However, the following issues still warrant discussion:(1) Since 2019, SE as a mode of transportation has gained attention, but the amount of research on it remains limited. Despite the many similarities between bike-sharing and SE, the differences in influencing factors and their impact directions necessitate a re-examination of these research findings. For instance, SE is larger and faster in speed, which may lead to higher expectations for riding infrastructure [5, 6]. Additionally, considerations such as fitness and the range of connectivity might cause changes in user groups and scenarios [7]. Therefore, can the research conclusions on bike-sharing be applied to SE? (2) There are many factors influencing shared e-scooters transferring to the metro (hereafter referred to as SE-metro transfer mode). However, survey questionnaires are constrained by participant acceptance and survey costs. How can we simplify the questionnaire to obtain an appropriate measurement scale? (3) The existing literature usually adopts the Theory of Planned Behavior (TPB) to analyze travel behavior, but it usually describes rational and mature travel decisions. When a new transportation mode is still in the promotion period and associated with another transportation mode, there is a sense of unknown to the public. Shared E-scooters System (hereafter referred to as SES) should consider its acceptability first. Is there a more appropriate theory? (4) Well-considered models are theoretically compelling but are often overly complex, leading to poor utility and statistical issues. Moreover, maintaining a certain amount of expansion flexibility to the model is often needed to suit the inclusion of different people's other concerns. What modeling concepts make more sense? (5) Weather factors are crucial for riding, and previous literature has conducted quantitative analyses on aspects such as precipitation, temperature, and air quality. However, despite the technical feasibility, more literature has yet to explore this from the subjective perspective of cyclists. In reality, people's experiences and decisions differ under the same climatic parameters; unless it's extreme weather, people's travel decisions in various weather conditions are often vague, making them suitable for subjective judgment.

This study primarily focuses on the influence mechanisms of the built environment, travel context, personal social attributes, and weather tolerance on the intention and attitude preferences towards using the "Shared E-scooters + metro" travel mode. We aim to efficiently simplify the assessment model's structure to provide references for professionals designing travel systems, policymakers, and government officials involved in infrastructure development. Firstly, we established a model to evaluate the satisfaction and loyalty towards the SE-metro transfer mode based on the Technology Acceptance Model (TAM), considering the extensibility and adaptability of application scenarios. We compared various alternatives from the perspectives of simplicity and goodness of fit. Secondly, we developed a survey scale to assess the intention of choosing SE for transfers, based on well-established research literature and scales, and proposed methods to simplify the questionnaire. Subsequently, we innovatively added the variable of subjective weather perception to comprehensively expand the model with the optimal fit among the alternatives, ultimately proposing a simplified version of the SE transfer choice model, M4.To achieve this, we conducted a questionnaire survey with 683 participants

from Changsha, China. Since this study is based on data from a single city, the conclusions should be cautiously generalized to other cities.

The structure of this paper is as follows. Section 2 reviews the research literature, including various factors influencing the use of bike-sharing or SE, as well as basic modeling theories and statistical methods. Section 3 introduces the SE in Changsha, China (hereafter referred to as CSSE), which is the case study of this paper. Section 4 describes the survey data using descriptive statistics. It determines a simplified version of the questionnaire for the SE-metro transfer mode through the calculation of the TAM conceptual model. Additionally, comparing the discriminant validity results of the TPB model validates the applicability and value of TAM as the research method in this paper. Subsequently, in Section 5, a new indicator of weather tolerance is added. The significant advantages of the expanded model are evaluated from the perspective of fit and explanation. Sections 6 and 7 discuss the influence mechanisms of various sample groups and different psychological variables. Finally, relying on the research conclusions, suggestions and measures for subsequent research and policy practice are summarized.

## 2 Literature review

This section presents the main findings of the research on shared cycle services. The first subsection focuses on the factors that influence the choice of riding trips, and the second subsection focuses on the theory and methodology of the quantitative study.

### 2.1 Factors influencing the choice of riding trips

When travelers make travel choices, there are significant differences in individual attributes and preferences, travel environment and other factors are also influenced. Reviewing the relevant literature on travel mode choice, we have organized the research results of various influencing factors into the following five categories: (1) Individual socio-economic attributes, (2) Travel background, (3) Built environment, (4) Attitudes and preferences and (5) Weather as shown in Table 1.

**2.1.1 Individual socio-economic attributes.** In terms of individual and household socio-economic attributes, gender, age, income, and education are all taken into consideration. Regarding gender, males exhibit a higher willingness to use non-motorized transportation and greater tolerance for its drawbacks [11]; however, females demonstrate stronger robustness to economic factors [13], and studies have shown that they are more accepting of transfer modes of transportation than males [8]. The main concerns for females are safety-related [47], and there is a lack of consideration for gender differences in various aspects of transportation, resulting in inherent gender barriers [10]. Young people have a more positive attitude toward bike-sharing [12], with affordability being an important factor [20]; however, they are also the group least willing to transfer multiple times [25]. Middle-aged individuals have more difficulty accepting bike-sharing as a new phenomenon [10], but there is still potential to be tapped into; they dislike long walks as a mode of transfer and are very sensitive to travel costs [8]. The biking rate among the elderly is very low, with the comfort and safety of biking facilities being their top concerns [11]. Income and education are negative factors for low-carbon travel modes [8], and some scholars believe that the positive correlation between income and education levels makes the preferences of these two groups tend to be similar [48], with electric bikes being more easily accepted by those with higher income and education levels [13].

**2.1.2 Travel background.** Whether for leisure or commuting, people are willing to use bike-sharing or SE for their trips [19, 21]. However, the number of transfers has a significant negative impact on this mode of travel [26]. Non-motorized travel is generally affected by

**Table 1. Literature summary of influencing factors.**

| Category | Factors | Impact on riding | |
|---|---|---|---|
| | | **Positive** | **Negative** |
| Individual socioeconomic attributes | Gender | [8, 9] | [10, 11] |
| | Age | | [9, 11, 12] |
| | Income | [13] | [12, 14, 15] |
| | Qualifications | [10, 16] | [11] |
| Travel background | Travel cost | [13] | [14, 17–20] |
| | Travel time | | [10, 11, 20] |
| | Recreational motives | [5, 12, 21, 22] | [9, 11, 23] |
| | Number of transfers | | [24–26] |
| | Travel distance | [7] | [5, 10, 11, 14, 23] |
| Built environment | Facility environment/ Land use | [5, 6, 23, 27–29] | [30, 31] |
| | Station distribution | [12, 19, 25, 31] | |
| | Dedicated road for riding | [7, 9, 10, 21, 27, 32] | |
| | Open green space | [19, 27, 32] | [12] |
| Attitudes and preferences | Sustainability and low-carbon awareness | [17, 33, 34] | |
| | Flexibility and Convenience | | [34] |
| | Comfortableness | [20, 35] | [34] |
| | Perceived Security | [14, 29, 36, 37] | |
| | Cautious approach to risk | | [6, 38, 39] |
| | Satisfaction | [17, 40, 41] | |
| | Social benefit | [17, 34] | |
| Weather | Quantity of rainfall | | [14, 19, 23, 31, 42–45] |
| | Temperature | [42, 43, 45] | [14, 23, 42] |
| | Humidity and wind | [23] | [32, 43–45] |
| | Community tolerance level | [24, 46] | |

transfer distance, with a threshold of 3 km, showing a positive trend initially, followed by a negative one [17]. Some studies indicate that SE has a larger threshold than bike-sharing [13]. Travelers are more willing to walk for transfers within 200 meters, but this willingness drops to only 20% at 700 meters [17]. Travel cost is inversely related to the willingness to choose this mode of transportation [20].

**2.1.3 Built environment.** The quality of riding infrastructure is a primary concern for riders [19]. The continuity and exclusivity of bike lanes, as well as the coherence of the riding experience, are crucial factors [49]. High-intensity land development, mixed-use areas, and the density of points of interest (POI) also serve as significant positive influences [23, 26]. Conversely, damaged bike lanes and the lack of safety measures like barriers greatly reduce the willingness to ride [26], especially at night [6]. Additionally, appealing environments such as parks, lakes, and greenery strongly attract users [32].

**2.1.4 Attitudes and preferences.** Although users with a strong awareness of environmental protection and health tend to prioritize low-carbon transportation methods [34, 42], ongoing publicity can effectively enhance public awareness [17]. However, people are more sensitive to improvements in comfort and service levels [35, 38], and their need for safety outweighs the need for flexibility [37]. This is especially crucial for women [34].

**2.1.5 Weather.** Weather is a significant factor affecting riding, with outdoor modes like SE being more impacted than enclosed travel methods [50]. Pleasant weather can boost users' willingness to ride [14], while rain and snow have the most significant effects [51], causing up to 75% of rides to be canceled due to rain. Shared bike rides starting near metro stations are

more susceptible to rain, leading to a significant decrease in riding frequency and duration [43]. This impact is even greater for inexperienced riders, as rain increases the risk by 2.5 times and snow by 4 times [49]. However, this influence mainly comes from immediate weather conditions, with subsequent weather changes being less important [44]. Temperature is another critical factor, with both summer and winter being less ideal for riding, while humidity amplifies the impact of high temperatures in summer [32].

The effects of wind vary; it can alleviate heat and increase the willingness to ride in summer [23], with riders often increasing their speed to create a cooling breeze. In contrast, frequent strong winds in autumn negatively impact riding, with each unit increase in wind speed decreasing the likelihood of riding by 0.8% [44, 45].Different groups have varying degrees of tolerance to weather, influenced by travel mode, purpose, environmental attitudes, and travel habits. This heterogeneity means that the impact of weather on travel behavior varies widely [46]. Understanding this variability is crucial for studying travel choices. However, as Nordbakke and Olsen point out, there is limited research on "weather tolerance" and its heterogeneous effects from the perspective of user diversity [9, 46]. Investigating this can highlight the sensitivity to weather impacts and underscore the travel behavior tendencies of different groups, which is of significant importance [24].

## 2.2 Theory and methodology

**2.2.1 Behavioral theory.**   User satisfaction and loyalty are overall evaluation metrics formed by users during purchase and use, primarily used to assess user experience and measure usage intentions. Satisfaction with the SE-metro transfer mode refers to SE users' experiences compared to their expectations regarding the products, services, infrastructure, and environment involved in using this travel mode. It is an emotional response. Loyalty to the SE-metro transfer mode mainly reflects whether users continue to choose SE as a travel mode, which is a behavioral decision and an extension of satisfaction research.

Since travel mode choice is generally considered a rational behavior, current research in the travel field often uses the Theory of Planned Behavior (TPB) as the foundational theory for constructing user satisfaction models [52]. However, as an emerging travel mode, SE differs from more conventional travel modes like cars and buses and is still in the process of being gradually recognized and accepted by the public. Thus, TPB is not the optimal explanatory theory for SE. The Technology Acceptance Model (TAM), based on the Theory of Reasoned Action (TRA) and TPB, was first proposed by Davis in 1989. It is a theoretical model for explaining people's adoption behavior, acceptance level, or usage intentions of new things [53]. TAM has the advantages of a simplified structure, robust model, and strong explanatory power for user behavior, making it more suitable for explaining travel choice behaviors related to SE.

As research has progressed, scholars have continually refined the Technology Acceptance Model (TAM) from three perspectives: the model framework, internal model factors, and the introduction of new theories into the model. Consequently, TAM evolved into TAM2 [54] and TAM3 [55]. Firstly, this evolution led to both a simplification and expansion of model variables. While removing attitude as a classic model variable and diminishing the influence of subjective norms, researchers also began to recognize the impact of user characteristics, environmental factors, and temporal factors on technology acceptance [55]. Secondly, the focus of research subjects has gradually shifted to specific groups. The scope of investigation has extended from internal model factors to external influence variables and from the information domain to multidisciplinary fields such as social, management, economic, and decision sciences [56]. These advancements have significantly broadened the applicability of the model.

**Table 2. Summary of literature.**

| Study | Valid responses | Data collection | Methodology |
|---|---|---|---|
| Choice preferences for shared bike | | | |
| Fernandez-Heredia et al [30] | 2555 | RP survey | SEM |
| Campbell et al. [57] | 496 | SP survey | Multinomial Logit |
| Shen and Chang [56] | 300 | RP survey | DEM |
| Ye et al. [20] | 1024 | RP survey | Mix Logit |
| Ingvardson et al. [34] | 1097 | RP survey | HCM |
| Tzouras et al. [37] | 129 | SP survey | Linear Regression |
| Choice preferences for shared e-scooter | | | |
| Tuli et al. [19] | Application monitoring and feedback | GPS application | Radom-Effects Negative Binomial Logit |
| Xin et al. [48] | 1023 | SP survey | Linear Regression |
| Weschke et al. [58] | 3834 | SP survey | Multinomial Logit |
| Carroll [13] | 431 | RP survey | Binary Logit |
| Li et al [59] | 830 | RP survey | Multinomial Logit |
| Shared bike connects to public transportation for interchange behavior | | | |
| Ai et al. [35] | Application monitoring and feedback | GPS application | Fuzzy Linear Regression |
| Bi et al. [23] | Application monitoring and feedback | GPS application | Binary Logit |
| Gan et al. [21] | 748 | RP survey | Random parameter Tobit and Multinomial Logit |
| Liu et al. [8] | 9.1 million | Smart Card Data | Binomial Logit |
| Guo et al. [17] | 415 | SP survey | HCM |
| Influences on riding in a given scenario | | | |
| Abou-Zeid and Ben-Akiva [41] | 594 | RP survey | HCM |
| Gebhart and Noland [43] | 1.3million | Information Disclosure Website | Negative Binomial Regression |
| Kaplan et al. [52] | 655 | RP survey | SEM |
| Zhao et al. [44] | Data monitoring and feedback | Automatic counters | Comparative and Residual Regression Analysis |
| Acharjee and Sarkar [29] | 815 | SP & RP survey | Multiple Indicators Multiple Causes (MIMIC) |
| Zhou et al. [31] | Application monitoring and feedback | GPS application | Geographically and Temporally Weighted Regression (GTWR) |
| Rashidi et al. [36] | 345 | RP survey | SEM |

**2.2.2 Methodological approach.** The commonly used statistical analysis methods in existing travel mode choice research include the Discrete Choice Model (DCM), which primarily aims at prediction; the Structural Equation Model (SEM), which primarily aims at explanation; and the Hybrid Choice Model (HCM) [35, 43, 44]. Our focus is on the intention to choose non-motorized transportation, particularly in the context of its integration with public transit. Relevant literature is summarized in Table 2.

The Discrete Choice Model (DCM) is a non-aggregate model based on utility theory, which expresses travelers' preferences for travel modes through utility value functions. It typically employs the principle of Random Utility Maximization (RUM), assuming complete rationality and commonly used methods include the Logit model and Probit model [60], with the Logit model being the most widely used. However, DCM generally assumes that the choice attitudes with individual heterogeneity are stable and consistent [38], and it explores less the impact of traveler heterogeneity [56], making it difficult to explain psychological factors such as individual choice preferences significantly related to behavioral intentions [61]. To address this issue,

SEM is introduced. SEM typically incorporates unobservable psychological factors such as attitudes and perceptions into the structural model in the form of latent variables and quantifies them through measurement models in the form of questionnaire items. It is widely used in areas such as travel satisfaction and loyalty, increasing riding rates [36], and differences in policy influence [59], with overall robust research conclusions [62] but with fewer studies on attitudes towards transfer travel. The Hybrid Choice Model (HCM) is a combination of both models, but the increased complexity of the model limits its practicality.

## 3 Case study background: SE in Changsha, China

CSSE's development went through a process of "barbaric growth, strict management, and standardized development." SE was first introduced to the Changsha market in 2018, accompanied by the frantic vehicle deployment by 10 operating companies to compete in the market. The number of CSSE peaked at 460,000 in November 2020, but only 60,000 were licensed, leading to a series of problems such as occupying lanes and chaotic parking, which raised concerns among citizens. In early 2021, due to safety and order considerations, almost all SE were recalled, and operations were halted. The recalled vehicles were either transferred to other cities for continued operation or disposed of, recycled, or entered the second-hand market. In 2022, after extensive discussions among the government, companies, and citizens, 50,000 SE from 8 operators were reintroduced to the market, with plans to gradually increase the number of vehicles but keep the total below 100,000.

The CSSE has an average riding time of 12.5 minutes and an average speed of 11.9 km/h. The average riding distance is 2.05 km, with 28% of rides being 0–1 km, 34% being 1–2 km, 16% being 2–3 km, and 21% being over 3 km. The commuting population accounts for 39% of users, and there is a strong coupling between order intensity and the distribution of workplaces and residences. Orders are evenly distributed among areas with high, medium, and low rent levels, with each accounting for 20%, 40%, and 40% of total orders, respectively. This provides equal opportunities for travel services for different income groups. However, high-intensity travel is relatively concentrated in areas with medium and low rents, benefiting difficult groups such as rural migrants and zero-employment households. Nighttime riding accounts for 12% of total rides, which is relatively high nationally, effectively promoting the city's nighttime economy and expanding the reach of the catering industry. Weather temperature is an important factor affecting SE usage, with ridership in spring and autumn being twice that of winter. Active users can reduce carbon emissions by an average of 40 kg per year, indicating that this mode of transportation is an effective low-carbon travel option.

The average riding distance for SE-metro transfer mode to metro stations is 1.9 km, and the order volume within 100 meters of the station is 2.2 times the average order volume. According to the experience of Chinese cities, the total length of the rail transit network is proportional to the order volume around the stations. Currently, Changsha has an operating mileage of 210 km and a planned total length of 456 km. Referring to the experience of similar Chinese cities, the final order ratio could reach around 6 times, indicating that Changsha's SE-metro transfer mode still has significant room for growth. Furthermore, currently only 51% of metro stations in Changsha have SE operating services, mainly because many peripheral metro stations are beyond the operating range of SE. However, the metro network in Changsha is basically structured as "sparse outside and dense inside," with low density and insufficient coverage of peripheral rail networks. This makes it the optimal market for the "SE + metro transit" mode of travel, which can significantly increase the population covered by metro services and attract more passengers. In addition to the SE-metro transfer mode, SE also plays a significant role in filling the gap in short- to medium-distance public transit services in some

areas. Currently, community buses, feeder buses, and shuttle buses are underdeveloped, with small passenger volumes and low economic benefits. SE, being flexible and convenient, has advantages in travel distances of 2–4 km, helping to make up for the lack of facilities in older communities, improving the convenience of life services, and shaping a "riding+" lifestyle circle.

## 4 Methodology

### 4.1 Conceptual model

This article intends to discuss the factors influencing the evaluation and intention to use the SE-metro transfer mode based on TAM and extend the model to include a latent variable for subjective perceptions of weather. Considering that the expanded model will have more mediating variables, making the model more complex and affecting its robustness, this article aims to construct and compare three TAM models based on the simplicity of the model without exceeding the complexity of the classic TAM model, as shown in Fig 1. This serves as the basis for studying the impact of weather factors.

(1) M1 Model. M1 is a simplified model of TAM after removing the attitude latent variable, as shown in Fig 1(A). Many studies have shown [53] that the attitude latent variable is only necessary for very novel and unfamiliar technologies, and in most cases, attitude and satisfaction can be combined [63]. Perceived Usefulness (PU) and Perceived Ease of Use (PEOU) are important antecedents of user satisfaction [53]. PEOU enhances users' sense of control, while PU enhances their sense of utility, both of which can reduce the gap between users' psychological expectations and actual perceptions, thereby increasing satisfaction. The stronger the PEOU, the easier it is for users to achieve their usage goals and enhance the pleasure of use.

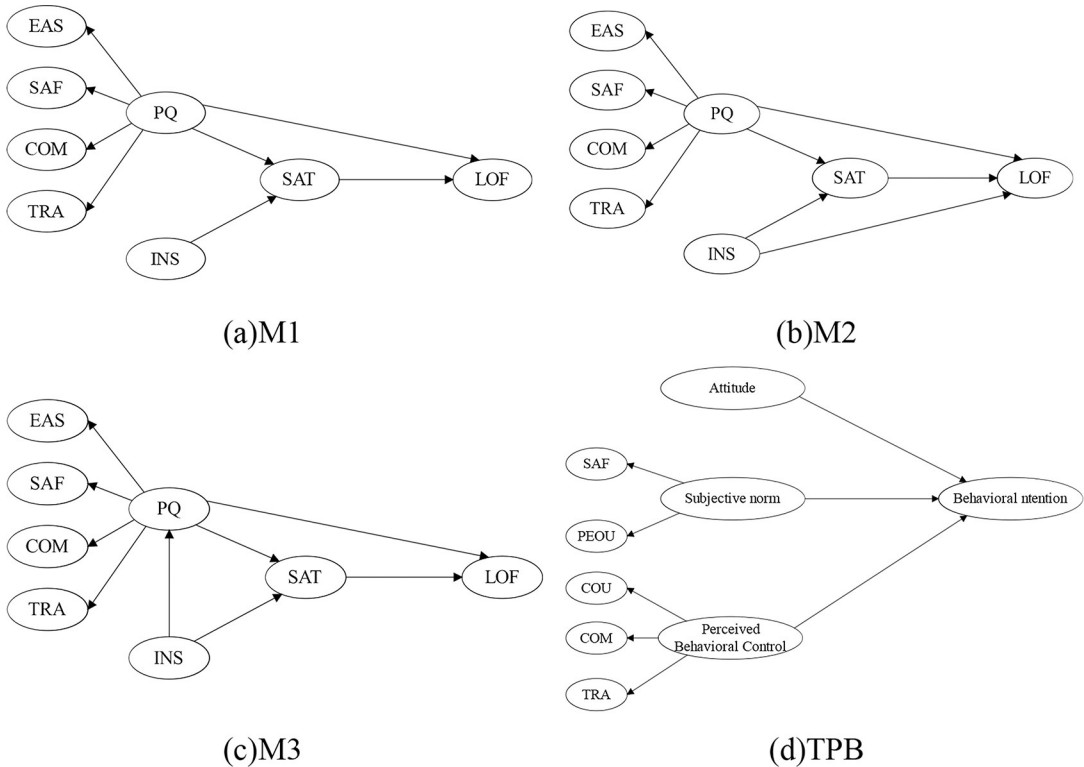

**Fig 1. Schematic diagram of the conceptual model.**

Thus, PEOU has a positive effect on PU [64]. Domestic and foreign studies have shown that satisfaction is a key driver of loyalty [65]. Loyalty includes attitude loyalty and behavioral loyalty, but due to the instability of measuring behavioral loyalty, more research focuses on attitude loyalty. Since the meaning of usage intention is consistent with attitude loyalty, PU also drives loyalty. By comparing the research results related to perceived service quality and based on the widely used SERVQUAL model, the model's PU corresponds to four second-order latent variables [66], namely Convenience of use (COU), Perceived safety (SAF), Comfort (COM), and Ease of transfer (TRA). During the ride-hailing process, users' understanding of PEOU mainly focuses on whether the facilities during the ride are user-friendly rather than whether the non-shared e-scooter system is easy to use. Therefore, this paper focuses on the facility conditions during the ride in the PEOU latent variable.

(2) M2 Model. Davis et al. [53] found that as experience increases, perceived ease of use will no longer be a significant variable affecting satisfaction, assuming that it affects satisfaction through PU unnecessary. This is more evident for short-distance commuting using the SE-metro transfer mode for daily commuting and schooling. After a period of riding, users can remain calm even in the face of poor riding facilities. Based on the principle of model simplicity, this paper cancels the assumption that ease of use affects usefulness in the M1 model and proposes the M2 model, as shown in Fig 1(B).

(3) M3 Model. Studies have shown that although the sensitivity of users to satisfaction decreases with an increase in riding experience, the objective impact of ease of use on users' experience does not change. Poor user experience can still reduce user loyalty. Therefore, this paper adds the impact of ease of use on loyalty to the M2 model and proposes the M3 model, as shown in Fig 1(C).

(4) TPB Comparative Model. To validate a behavior selection model suitable for SE travel, the TPB model is considered as an alternative. By comparing the model's explanatory power, the optimal model with universal research value is selected. Fig 1(D) of the TPB model discusses behavioral intentions based on three variables: attitude toward behavior, subjective norms, and perceived behavioral control. Attitude toward behavior refers to users' evaluation of specific behavior, i.e., user satisfaction; subjective norms refer to the social pressure individuals feel about whether to adopt the behavior; in the context of SE, it refers to the physical traffic environment rather than others' evaluations, specifically referring to social norm culture (SAF) such as road order and potential interference from transportation facilities themselves (PEOU); perceived behavioral control refers to the degree of control users have over the behavior. The ease of use of SES (COU), riding comfort (COM), and the reliability of SE transfer functions (TRA) are highly correlated with the former, indicating a common essence. Therefore, these three variables are extracted as second-order factors of perceived behavioral control. Individuals form a behavioral inclination after integrating attitudes toward behavior, subjective norms, and perceived behavioral control, ultimately forming loyalty to SE transfers.

(5) M4 with extending weather perception dimensions

Research has shown that the three factors influencing riding behavior most are terrain, weather, and safety [24, 29]. SE largely eliminates the influence of terrain, and there has been more research on riding safety [67]. However, few have focused on how SE impact on weather perception. Studying the common extension pathways of TAM [68], we propose to compare M1-M3 models and then incorporate weather perception as a latent variable into the optimal one. The previous survey showed that users usually do not switch their dissatisfaction with the weather to SES but will reconsider using SE. It is also believed that severe weather does not change the value of SE usage, but it can objectively create barriers to it. Therefore, extended model M4 proposes introducing the weather dimension and hypothesizes it significantly affects PEOU and LOF.

## 4.2 Data

The survey data was collected from a questionnaire survey on the SE-metro transfer mode conducted in Changsha, Hunan Province, on June 24–25, 2021. The measurements used a five-point Likert scale. Before the formal survey, a preliminary survey was conducted around 30 metro stations with high SE demand. The areas around these preliminary survey sites included residential, commercial, office, school, transportation hubs, and park landmarks. By excluding unsuitable locations for conducting the questionnaire, 20 formal survey sites were ultimately selected.

The survey used an intercept method to interview nearby residents randomly. Respondents signed a written informed consent form before filling out the questionnaire. Minor respondents completed the survey with the consent of their guardians. To avoid volunteers' subjective bias towards certain respondent groups, stratified sampling was used based on user characteristic data released by the SE company, controlling for age, gender, and other conditions. The answering time was controlled between 10–20 minutes. With the respondents' consent, a total of 1012 questionnaires were distributed, and 875 were collected. After data cleaning and validity checking, 683 valid questionnaires were obtained, with an effective rate of 78.1%. The questionnaire utilized in this study is delineated in the (S2 File). The data amassed and a synopsis of the scores by dimension are presented in (S1 Data) and (S1 File), respectively.

The descriptive statistics of the respondents' data in Fig 2 show that the gender distribution of the user group is relatively balanced, generally younger and more educated. College students, freelancers, and enterprise employees are in the top three, and the large proportion of middle- and low-income users reflects the occupation distribution. The distribution of riding time and distance indicates that SE is mainly used for short-distance travel, but it is outsides the comfort range of walking. However, the proportion of riding distance below 1km and 2-4km is not low, indicating that SE has strong competitiveness in the face of walking and short- and medium-range public transportation, and its scope of application is broader. Analyzing the travel time of the metro part of its transfer, the travel time of more than 15 minutes accounts for 75%, corresponding to the travel distance of more than 10km, indicating that the metropolis can better reflect the value of SE's transfer. In terms of riding frequency, most people only use SE as an alternative travel mode, which indicates that there is still much room for growth in the commuting field.

It can be observed that the composition of the survey respondents (such as gender ratio, age distribution, education level, etc.) corresponds to the user data disclosed in market reports [69], reflecting the current status of the SE market composition and the natural selection preference of a certain specific group of people for SE. This further reveals the applicability and value of this study.

### 4.3 Computational procedure

**4.3.1 Exploratory factor analysis.** The main purpose of EFA is to identify whether the latent variables align with the predetermined items in the conceptual model and to reduce the observed items while maintaining the model's explanatory power. The study employed the Partial Least Squares method to test 32 items, of which COU5, TRA2, PEOU5, and LOF4 were excluded due to factor loadings less than 0.7. Consequently, 28 items were retained. The KMO value after excluding items was 0.877, indicating that the sampling was adequate. Bartlett's test of sphericity was significant, and the rotated component matrix based on the maximum variance method showed that the absolute values of all elements exceeded 0.5. The questionnaire items could be divided into 8 different latent variables, namely: (1) Convenience of Use (COU). Includes items like "neat placement of vehicles" and "easy to scan and rent a vehicle";

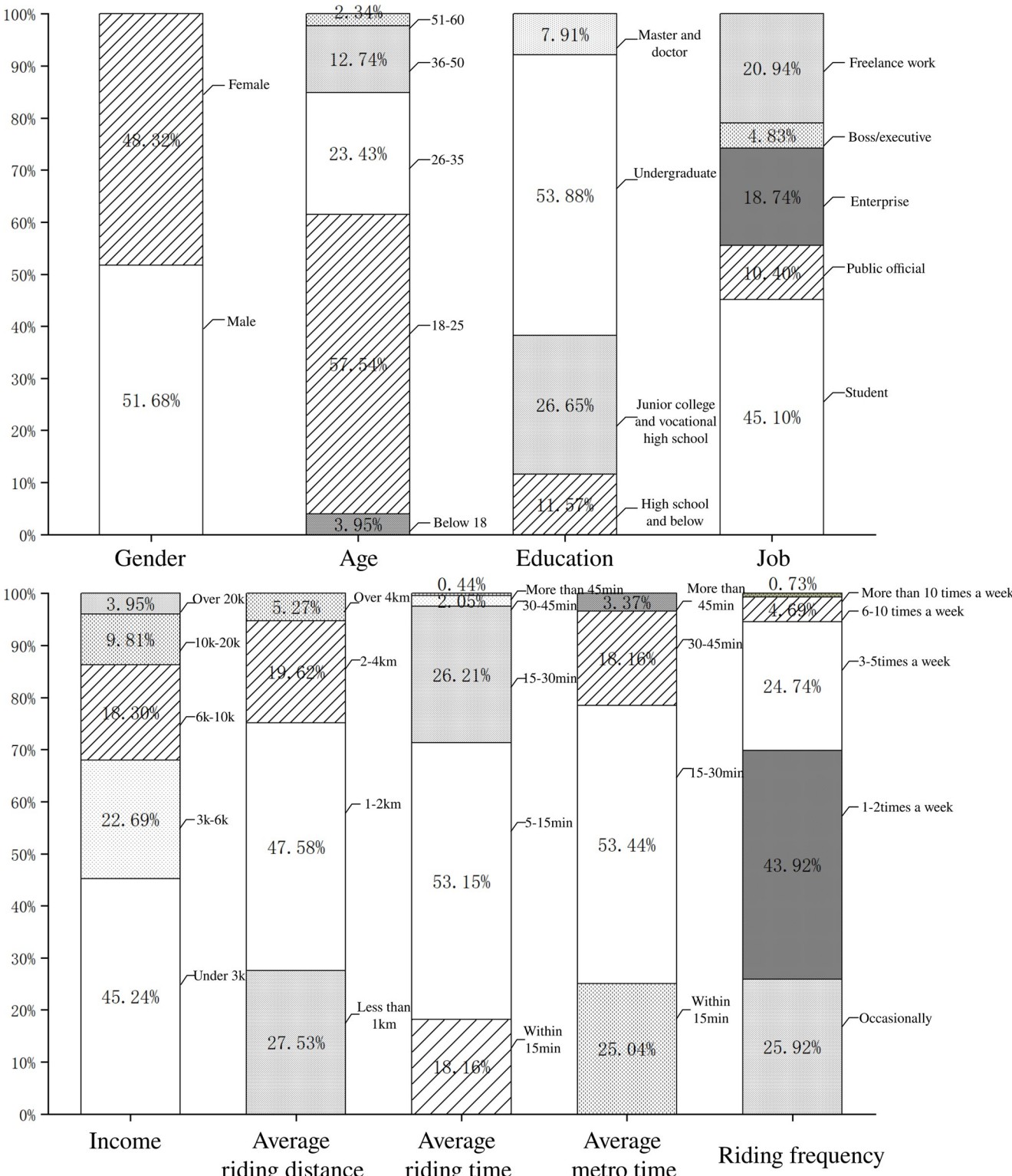

**Fig 2. Descriptive of sample characteristics.**

(2) Perceived Safety (SAF). Includes items like "good traffic order" and "few interferences while riding"; (3) Comfort (COM). Includes items mainly related to the riding comfort of the e-scooter.; (4) Ease of Transfer (TRA). Defined by items such as "reasonable layout of metro entrances and exits," "more convenient than bus transfers," and "close proximity of pickup and drop-off points."; (5) Perceived Ease of Use (PEOU). Represents indicators related to perceived ease of use, such as "wide enough bike lanes" and "shaded paths."; (6) Weather Preference (WEA). Measured by different respondents' tolerance to four weather conditions: "haze, light rain, heat, and cold."; (7) User Satisfaction (SAT). Composed of various evaluation factors such as "willingness to recommend to others" and "se has improved the quality of life."; (8) User Loyalty (LOF). Measured by users' choice of se as a transfer mode in various hypothetical scenarios.

**4.3.2 Simplification of questionnaire items.** We aim to use the PLS method to select the best-fitting model among M1~M3 of different questionnaire sizes based on the streamlining principle. $R^2$ and BIC values are both indexes for determining model fit, but $R^2$ tends to favor models with a larger number of variables, which is a drawback that can lead to unnecessarily complex models. At the same time, BIC is based on Bayesian Information Theory, which better represents the overall best goodness-of-fit. Distinguishing validity is a crucial criterion for determining whether a questionnaire is refined. HTMT (heterotrait-monotrait ratio) represents the ratio of between-trait correlation to within-trait correlation and is an indicator for assessing discriminant validity. An HTMT can be calculated between traits (i.e., latent variables) higher than 0.90, meaning poor discriminant validity [70]. In this study, HTMT was selected as the basis for simplifying questionnaire items, while BIC was used as the discriminant indicator for model comparison.

Deletions for streamlining are made in order of magnitude based on their model's explanatory power. PU is jointly reflected by four second-order latent variables (COU, SAF, COM, and TRA), which are conceptually easily cross-repeated. In this paper, the simplification started from this point onwards, and we used the factor loading to measure their contribution's magnitude. The questionnaire items screened by EFA were used as the Base group. Based on the principles of single variable control and not missing any possibility, we started from COU4 (0.739) and SAF4 (0.756), which have the smallest factor loading, respectively. The items were divided into two groups to list all the deletion schemes stepwise incrementally and then terminated when each latent variable's observation variables were reduced to three. Finally, a total of 10 kinds of schemes were taken into account, as shown in Table 3. Except for schemes 8 and 9, which were eliminated due to not meeting the requirements of HTMT values, the values of the remaining schemes were relatively close, indicating that the model generally conforms to statistical criteria. In the most simplified schemes, 5 and 10, the former obtained a smaller BIC value, resulting in a final questionnaire with 23 items. This paper investigates how to improve user satisfaction and loyalty, combining BIC values of Base and 10 schemes with the principle of simplicity. It can be seen that M2 is the optimal model, as shown in Table 3. Therefore, this paper selects "M2+Scheme 5" as the optimal model for the following classification comparison and as the base model of M4.

**4.3.3 Applicability analysis of research methods.** For the TPB model, another alternative research method, when calculating the discriminant validity, it was found that the second-order factors PEOU and SAF, which constitute the 'subjective norms' variable, do not meet the requirements. The correlation values of 0.822 and 0.873 in Table 4 far exceed the square root of the AVE value of this variable, which is 0.786. This indicates that the model structure of TPB cannot adapt to the psychological motives of users under the conditions of the SE-metro transfer mode. At the same time, the second-order latent variables COU, COM, and TRA, which constitute the 'perceived behavioral control,' are all close to the critical value (0.665,

**Table 3. Comparison of model fitting and question item screening results.**

| Groups | Program | Excluding items | HTMT | | Latent variable | BIC | | |
|---|---|---|---|---|---|---|---|---|
| | | | PU-SAF | PU-COM | | M1 | M2 | M3 |
| Base | | COU5, PEOU5、LOF4, TRA2 | 0.869 | 0.866 | LOF | -169.077 | -168.990 | -196.561 |
| | | | | | SAT | -154.616 | -156.280 | -156.239 |
| 1 | 1 | COU4 | 0.893 | 0.875 | LOF | -169.273 | -169.211 | -196.516 |
| | | | | | SAT | -602.291 | -152.465 | -152.419 |
| | 2 | COU4, SAF1 | 0.858 | 0.896 | LOF | -168.682 | -168.577 | -196.397 |
| | | | | | SAT | -500.718 | -153.436 | -153.395 |
| | 3 | COU4, SAF1, COM4 | 0.877 | 0.849 | LOF | -169.353 | -169.312 | -196.205 |
| | | | | | SAT | -527.424 | -149.446 | -149.388 |
| | 4 | COU4, SAF1, COM4, TRA1 | 0.883 | 0.863 | LOF | -169.233 | -169.234 | -195.982 |
| | | | | | SAT | -584.850 | -148.847 | -148.805 |
| | 5 | COU4, SAF1, COM4, TRA1, SAF4 | 0.879 | 0.863 | LOF | -168.963 | -168.903 | -196.144 |
| | | | | | SAT | -498.682 | -149.853 | -149.802 |
| 2 | 6 | SAF4 | 0.854 | 0.880 | LOF | -168.583 | -168.451 | -196.584 |
| | | | | | SAT | -501.363 | -157.212 | -157.161 |
| | 7 | SAF4, COU4 | 0.878 | 0.893 | LOF | -168.963 | -168.858 | -196.626 |
| | | | | | SAT | -534.982 | -153.382 | -153.327 |
| | 8 | SAF4, COU4, TRA1 | **0.904** | **0.908** | LOF | -168.830 | -168.746 | -196.444 |
| | | | | | SAT | -581.791 | -153.062 | -153.023 |
| | 9 | SAF4, COU4, TRA1, COM4 | **0.925** | 0.859 | LOF | -169.667 | -169.661 | -196.309 |
| | | | | | SAT | -616.566 | -148.965 | -148.910 |
| | 10 | SAF4, COU4, TRA1, COM4, SAF5 | 0.877 | 0.882 | LOF | -170.239 | -170.120 | -198.398 |
| | | | | | SAT | -470.851 | -151.911 | -151.795 |

0.762, and 0.748), further proving that the TAM model is the appropriate research method for SE transfer travel.

## 5 Results

The results of the questionnaire survey in this study are shown in Fig 3. Among the measurement results of 8 latent variables, users have a higher recognition of the convenience of using SE (COU: 3.694), while their evaluations of security (2.901), loyalty (2.795), and weather perception (2.346) are lower. At the same time, the remaining first-order latent variables, COM (3.398) and TRA (3.147), as well as PEOU (3.149) and SAT (3.438), are all between 3.1 and 3.4, indicating an overall weak favorability among users. The degree of dispersion of scores for each latent variable is relatively high, with standard deviations concentrated in the range of 0.7 to 0.9. Descriptive statistics indicate that the current status of SE as a mode of transportation still needs to be solid, and there is still considerable room for improvement in vehicle maintenance. Even minor disturbances could change user choices. Additionally, respondents have heterogeneous evaluations of transfer service functions, indicating a long road ahead for development.

**Table 4. Discriminant validity of the TPB comparative model.**

| Latent variable | COM | COU | PEOU | SAF | TRA | Subjective norm | Perceived Behavioral Control |
|---|---|---|---|---|---|---|---|
| Subjective norm | 0.405 | 0.279 | **0.822** | **0.873** | 0.379 | 0.786 | - |
| Perceived Behavioral Control | 0.762 | 0.665 | 0.412 | 0.424 | 0.748 | 0.492 | 0.795 |

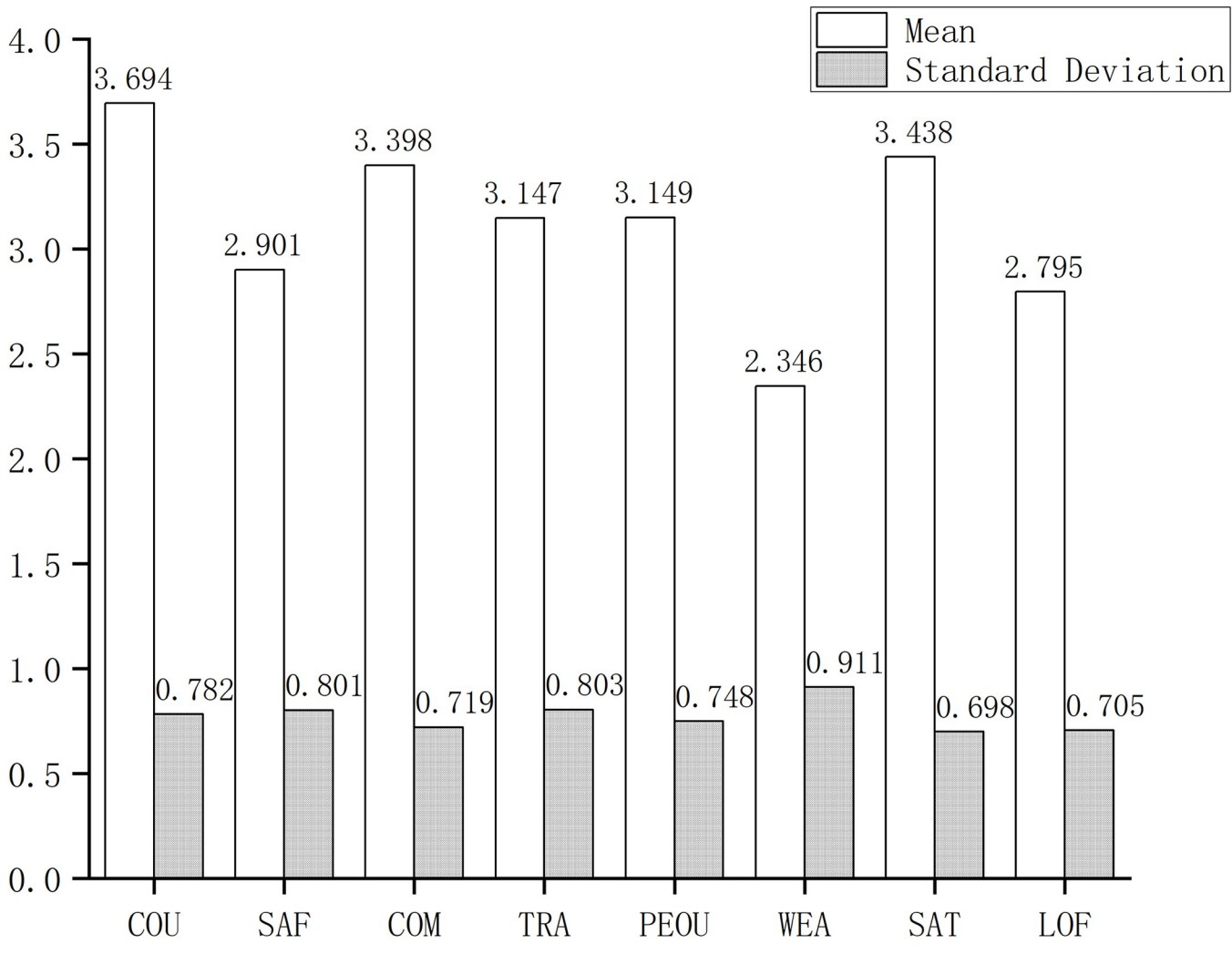

**Fig 3. Scores for each dimension.**

### 5.1 M2 modeling results

The calculation results of model M2 are shown in Fig 4. The factor loadings between latent variables and observed variables are all greater than 0.70, and all path coefficients have p-values lower than 0.05, indicating that the model meets statistical requirements. Regarding the intermediate path results in Table 5, the path starting from PU has a VAF value of 25.76%, which falls within the range of 20% to 80% for partial mediation effects [71], indicating that improving the psychological ratings of PU will indirectly enhance the level of user loyalty to a certain extent.

### 5.2 M4 modeling results

Based on M2, M4 adds two assumptions about weather perception: (1) Weather perception will not affect the system's usefulness but its ease of use. (2) Users will not complain about the system because of the weather, but it will affect their loyalty. In addition to the path with a complete mediation effect, the two partially mediating paths shown in Table 5 have a starting point of WEA. This path is a special case of a negligible dual mediation path, with a VAF value of only 7.44%.

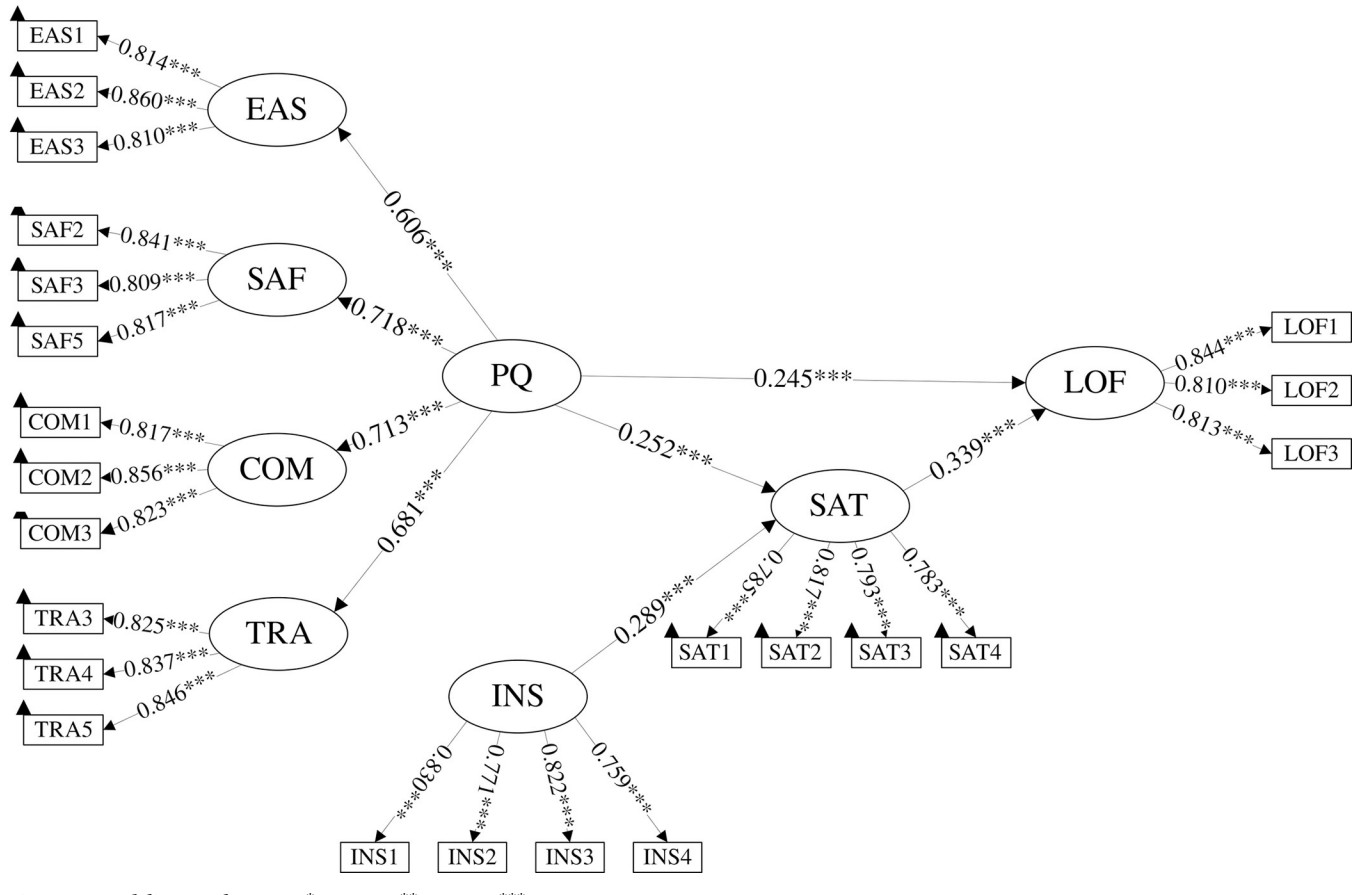

**Fig 4. M2 modeling results.** Note: * $p < 0.05$, **$p < 0.01$, *** $p < 0.001$.

The results of M4's reliability and factor correlations are shown in Table 6. The Cronbach's Alpha values meet the standard of being greater than 0.7, demonstrating the model's reliability. $Q^2$ is a measure of whether the endogenous structure has predictive relevance [72], and values greater than 0 indicate good predictive ability for the model [71]. On the right side are the discriminant validity results based on the Fornell-Larker criterion. The values on the diagonal of the matrix represent the square root of the Average Variance Extracted (AVE) for each variable, which is greater than the correlation coefficients with other constructs, demonstrating the model's satisfactory discriminant validity [73].

**Table 5. Mediated effects results of M2 and M4.**

| Model | Path | Indirect Effects | Confidence Interval | | P-Value | VAF | Mediation |
|---|---|---|---|---|---|---|---|
| | | | 2.50% | 97.50% | | | |
| M2 | PU-> SAT -> LOF | 0.085*** | 0.059 | 0.117 | 0.000 | 25.76% | Partial |
| M4 | PU-> SAT -> LOF | 0.084*** | 0.058 | 0.114 | 0.000 | 28.57% | Partial |
| | WEA ->PEOU-> SAT -> LOF | 0.016*** | 0.008 | 0.027 | 0.001 | 7.44% | None |

Note: *$p < 0.05$

**$p < 0.01$

***$p < 0.001$.

**Table 6. Relevance and reliability of M4.**

| Cronbach Alpha | Q² | Latent Variable | COM | COU | PEOU | LOF | PU | SAF | SAT | TRA | WEA |
|---|---|---|---|---|---|---|---|---|---|---|---|
| 0.778 | 0.377 | COM | 0.832 | | | | | | | | |
| 0.774 | 0.365 | COU | 0.244 | 0.828 | | | | | | | |
| 0.808 | 0.388 | PEOU | 0.303 | 0.262 | 0.796 | | | | | | |
| 0.762 | 0.350 | LOF | 0.218 | 0.282 | 0.419 | 0.823 | | | | | |
| 0.803 | 0.202 | PU | 0.713 | 0.606 | 0.487 | 0.378 | 0.830 | | | | |
| 0.761 | 0.348 | SAF | 0.359 | 0.250 | 0.439 | 0.280 | 0.718 | 0.822 | | | |
| 0.805 | 0.383 | SAT | 0.271 | 0.282 | 0.412 | 0.435 | 0.394 | 0.288 | 0.795 | | |
| 0.785 | 0.388 | TRA | 0.303 | 0.233 | 0.314 | 0.254 | 0.681 | 0.310 | 0.232 | 0.836 | |
| 0.804 | 0.383 | WEA | 0.129 | 0.074 | 0.164 | 0.273 | 0.189 | 0.165 | 0.103 | 0.140 | 0.794 |

## 5.3 Model selection

PLS modeling cannot optimize any global scalar function compared to maximum likelihood estimation, lacking indices that can provide users with global model validation [74]. However, GoF (Goodness of Fit) serves as a fitting index for PLS, which can compensate for this deficiency. The size of GoF is calculated according to Formula (1) using the corresponding values provided by the SmartPLS software. The GoF value for M2 is 0.344, indicating a moderate fit (0.25, 0.36) [74], while the GoF value for M4 is 0.362, indicating a high fit. This suggests that the M4 model, which includes the weather perception variable, has a better fit and stronger explanatory power.

$$GoF = \sqrt{\overline{Community} \times \overline{R^2}} \tag{1}$$

In the modeling calculations for both the test set and the validation set survey data, it was found that the M4 model maintained consistent usability and effectiveness across different samples. This demonstrates the universal applicability of the conclusions of this study.

## 6 Discussion

Fig 5 presents the estimation results of the M4 model, where different path coefficients represent the direct relationships between latent variables. Based on the evaluation results of the model, we discuss in detail the influences of sample characteristics and the utility of psychological variables.

### 6.1 Sample characteristic differences

Discussing the preference for the SE-metro transfer mode among different sample groups, divided by respondent gender: The M4 model maintains a relatively consistent explanatory power in both groups. Men show greater interest in emerging transfer modes, with measures to improve riding comfort being particularly effective for them. Women, on the other hand, are more concerned about their image. They are more selective about riding safety and road conditions and have a lower tolerance for weather conditions. They consider improving the evaluation of SE from aspects such as ensuring the smoothness and continuity of non-motorized lanes and providing raincoats.

According to the respondents' educational backgrounds, they were divided into two groups: high school and junior college, and bachelor's degree and above. The M4 model had a more prominent explanatory effect on users with a bachelor's degree and above. Perceived usefulness and satisfaction levels significantly influenced the loyalty of this group. Measures such

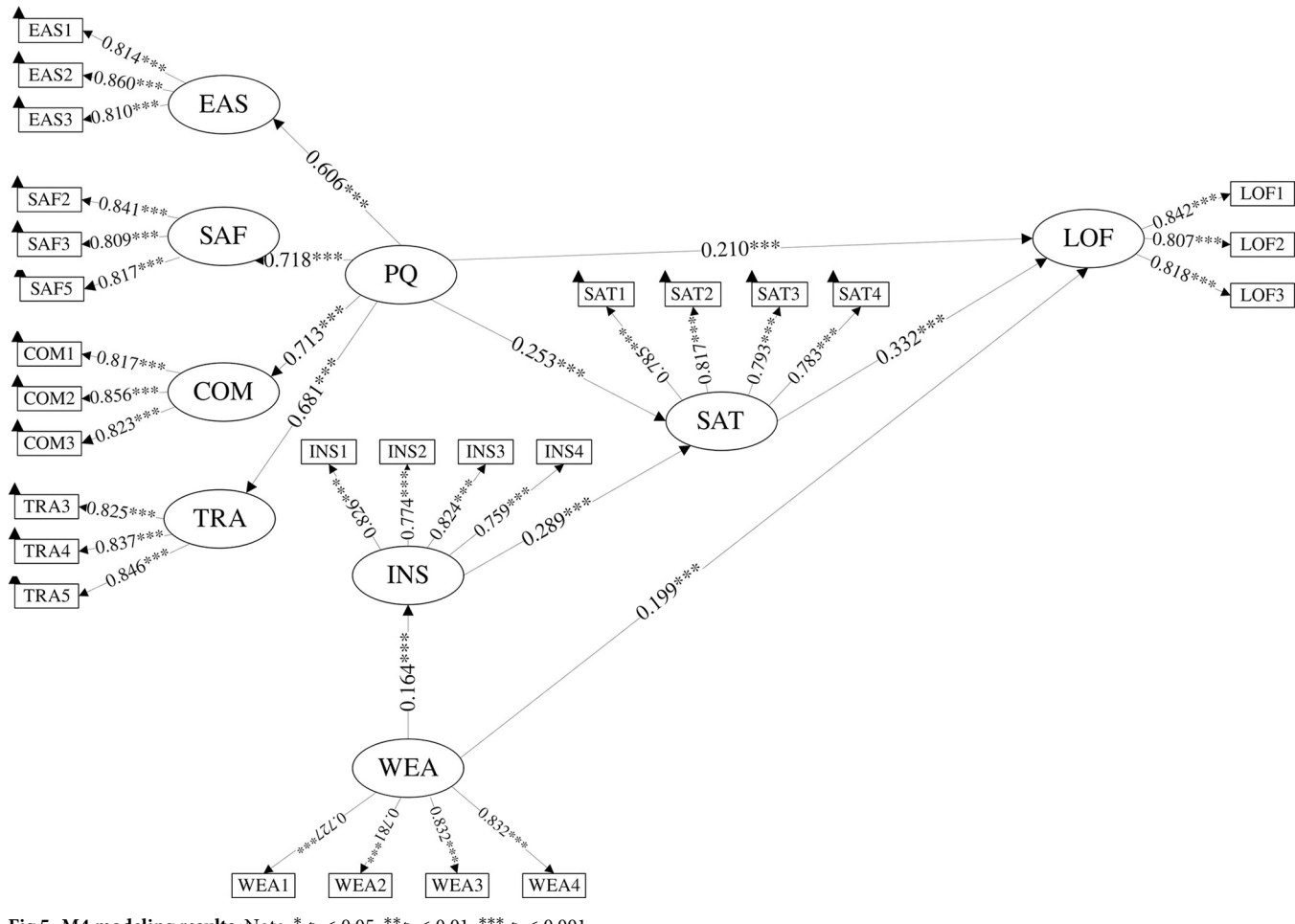

**Fig 5. M4 modeling results.** Note: * $p < 0.05$, ** $p < 0.01$, *** $p < 0.001$.

as simplifying system operations and enhancing the supply of e-scooters are recommended. On the other hand, those with a high school and junior college education focus more on individual circumstances. After comprehensive consideration of riding safety, comfort, and tolerance for weather, they make choices accordingly.

Using 25 years old as the cutoff, respondents were divided into two groups: Generation Z, representing young people entering society, and Millennials, representing stable working-age adults. Generation Z has a stronger acceptance of new things and focuses on the transfer function and reliability of SE. Millennials often commute by bike, and their travel is continuous and stable. They are more willing to make subsequent choices based on their own user experience, so the impact of weather factors is minimal. For older middle-aged citizens, a lack of understanding of SE and difficulty operating the software hinder them from trying such transfer modes. Improving ease of use can help SE gain greater popularity.

Users were grouped by occupation, with university students, executives, and freelancers characterized as having more freedom. At the same time, government and corporate employees with fixed working hours were classified as non-freedom groups. The freedom group has more leisure time, leading to higher demands for the quality of SE-metro transfer mode travel. They consider various influencing factors comprehensively. As for non-freedom groups like government departments and corporate employees, although they have stable travel

characteristics, fatigue from work makes them pay more attention to the riding experience. Dangerous road conditions and uncomfortable riding experiences greatly interfere with their choice of SE. Optimization can be achieved by separately isolating non-motorized lanes, maintaining traffic order, improving SE configuration, and riding power.

## 6.2 The influence of psychological variables on utility

**6.2.1 Weather tolerance.** The inclusion of weather tolerance helps enhance the model's explanatory power. This latent variable can directly influence PEOU and LOF. Despite many related studies, few have considered subjective weather perceptions as influencing variables. It is also important to design questionnaire items more specifically to prevent overlooking survey information such as light rain.

Improving perceived ease of use can offset the impact of bad weather. Different regions should have different strategies based on factors such as precipitation, economic conditions, temperature, and air quality. For example, rubber seat cushions without sponges can be used for SE to improve drainage and anti-freezing performance. Planting large canopy trees as roadside trees can effectively alleviate the impact of haze, light rain, heat, and cold weather while ensuring continuous riding. Additionally, continuous rain shelters can be set up on roads with heavy non-motorized traffic, a measure already applied in cities like Singapore and Hong Kong, China. Maintaining the riding quality and safety of roads, ensuring continuous riding environments, and increasing riding speed can also effectively reduce the impact of weather.

The significant path coefficient from weather to loyalty implies that people are easily inclined to change their travel modes due to weather conditions, and they are only willing to use riding modes like SE in very comfortable situations. The impact of weather on loyalty can be categorized as follows: First, weather can impact personal image. Variable weather conditions may cause users to sweat or get wet in the rain, damaging their image at work and in daily life. Therefore, providing locker rooms and shower facilities in office buildings can facilitate users in cleaning their clothes, effectively enhancing their resistance to the effects of weather [75]. Second, weather can impact work or travel plans. Previous studies have suggested that facilities such as locker rooms can help avoid the constraints of riding in formal attire, providing conditions for changing from sportswear to work clothes [19]. Third, weather can hinder the choice of users with special needs. Equipping the vehicle with a mini disposable raincoat box and a foldable storage box can meet the needs of sudden rain and carrying large items.

In the model results, the impact of weather on loyalty (LOF) is greater than that on perceived ease of use (PEOU), indicating that weather's influence on transfer choices is more emotional rather than a rational choice based on the real riding environment. It is worth noting that the time spent on transfer travel exposed to outdoor environments is short, and users' inquiries focus on their sensitivity to minor weather changes. Overall, the fluctuation is small, so the numerical values of the weather-related paths are not significant.

**6.2.2 Perceived Usefulness.** The significant path from Perceived Usefulness (PU) to Loyalty (LOF) implies that even under the assumption of suboptimal satisfaction, the construction department can still maintain users' loyalty to SE by improving PU. The importance ranking of the four latent variables comprising PU, namely SAF, COM, TRA, and COU, is consistent with expectations, with all of them receiving high weights. This is in line with previous findings in shared bicycle research, where users are more concerned about the safety of SE. This is mainly due to the generally poor traffic safety conditions and road infrastructure, as well as the higher speed and greater weight of SE compared to bicycles.

We must also consider the influence of some underlying factors. Although all respondents have experience with the SE-metro transfer mode, the transfer rail function is only part of the

purpose for choosing the SE. This may be an important reason for the relatively low TRA values.

**6.2.3 Perceived Ease of Use.** Perceived Ease of Use (PEOU) not only directly affects satisfaction but also indirectly influences users' loyalty to the SE-metro transfer mode. This indicates that the social reputation of SES is closely related to a good usage environment, including continuous and ample riding space, well-maintained road surfaces, shaded areas, and clear directional signs. Neglecting to optimize the riding environment may lead to the government making incorrect decisions regarding support for SES, consistent with a comparative study on the United States and Canada [76]. Most of the lower-rated items are related to government policies and infrastructure, indicating an urgent need for adjustment in Changsha's government policies related to SES development. This includes infrastructure construction, including transfer functions, which should listen more to the opinions of riders.

Previously, the ban on SES by the Changsha government was based on complaints from citizens. However, these complaints, while reflecting some actual issues with SES, mostly highlight the deficiencies in the ease of use of riding space and facilities, such as mixed traffic with motor vehicles and illegal parking. In fact, these issues are mostly caused by private car owners. Still, based on the principle of fairness, low-income groups have the right to equal road rights as private car owners. At the same time, the mismatch between the number of parking lots for motor vehicles and non-motor vehicles has caused problems with nowhere to park SE, which also deserves attention.

**6.2.4 User satisfaction.** Satisfaction is the most important psychological latent variable in the choice of SE, with few users giving extremely negative evaluations. They generally exhibit a more satisfied and concentrated attitude. Maintaining an affordable price is a decisive factor in satisfaction. Since the resumption of SE operations in Changsha, prices have doubled or even surpassed those of buses and subways. This indicates that there are many deficiencies in current operation management. Firstly, the policy of maintaining multiple operators and controlling the number of SEs put into operation has led to a lack of competition among operators. They are confident in maintaining high prices even when there is a high demand during peak hours to achieve higher profits. Secondly, the limited number of SE put into operation has led to high fixed costs for operators, which is also one of the reasons for the increase in prices.

Gradually increasing the number of operating vehicles and appropriately reducing the number of operators is key to future CSSE policies. The concept of low-carbon environmental protection is gradually becoming popular in China, which can promote the formation of satisfaction. However, as a developing country, Chinese people value the sense of honor represented by their means of transportation: using a car is considered more decent while using a non-motor vehicle is seen as a sign of a less successful career. This outdated notion is particularly evident among middle-aged and older people, which is also a major reason why SEs are difficult to promote. Therefore, it is important to reshape users' social values, establish a good social image of riding, and actively promote environmental values. A good measure is to recommend the use of SEs to friends around them. The power of mutual recognition may be more effective and lasting.

## 6.3 Scope of SE use

The emergence of SE has significantly increased users' tolerance for travel distances, making it suitable for a wider range of areas and travel purposes. The street block scale in the periphery of cities has gradually expanded with urban expansion, and the SE-metro transfer mode is more in line with the typical characteristics of low-density and super-large blocks than shared bicycles, effectively alleviating the negative impact of reduced road network density. At the

same time, for groups such as the elderly who lack physical fitness and do not like shared bicycles, or in cities with large terrain variations, SE has significant advantages. SE also features pedal functions, which can still meet fitness needs.

Various factors influence the choice of the "SE + metro" travel mode in work and life. For example, complex temporary situations such as picking up children or spouses, shopping, or frequent fieldwork are not suitable for this type of travel mode. However, in situations of traffic congestion and difficulty in parking, the choice of SE-metro transfer mode will be promoted.

## 6.4 Limitations and future research

This study has several limitations. Firstly, it is based on survey data from only one city. The geographical environment, urban planning, cultural background, and riding facilities vary greatly between different countries and even within different cities. Therefore, the study's conclusions can only be cautiously generalized to cities in southern China and some cities in Southeast Asia. Secondly, due to the reluctance of older and less educated groups to use SE, there is a slight shortage of samples compared to other categories. The promotion measures proposed in this paper may only be applicable to some residents. Thirdly, the complex characteristics of SE, combining "emerging" and "traditional," are not consistent with the TAM theory, which requires the research object to be completely innovative. Moreover, current research mainly focuses on the background where SE has yet to be widely deployed, and the choice of transfer travel may change further with the popularization of SE use. Fourthly, although psychological variables were used to explore some heterogeneity in SE travel choices, the handling of random preferences is still not satisfactory. At the same time, the research is based on the most concise form of practicality consideration of the model, but whether it meets the optimal explanation and prediction capability requires further research.

Future research directions mainly include four aspects. Firstly, establish a mixed-choice model (such as the UTAUT model), combine the theoretical framework of TAM for emerging things with the explanatory advantages of TPB for traditional travel modes, and comprehensively explore the heterogeneity of users' psychological choices of SE-metro transfer mode. Secondly, conduct targeted research on specific target groups such as the elderly or those with low education levels, tourists, etc., to clarify the key factors for market expansion. Thirdly, accumulate travel data of users over a long time and large span to carry out a time-series study under the conditions of SE market maturity. Fourthly, comparing the transfer functions of various shared mobility modes may reveal the applicable scenarios of other shared mobility modes. For new modes of transportation such as e-scooters that still need to be deployed in the Chinese market, comparative studies based on the Meta-analysis method can be considered.

## 7 Contributions and conclusions

This paper reveals user preferences for shared e-scooter systems and their riding environment in the context of the "Shared E-scooter + Metro" transfer mode. Our contributions to research, policy, and practice are as follows.

First, we included user perception of weather to understand and predict its impact. The model indicates that weather perception significantly affects perceived ease of use (PEOU) and loyalty. Features such as rubber seats and office building facilities like changing rooms, as well as well-designed road-riding spaces, play an important role in mitigating the negative effects of bad weather.

Second, our research shows that there are many areas for optimization in the current shared e-scooter policies in Changsha. It is crucial to increase the number of shared e-scooters, reduce

the number of operators appropriately, ensure riders' rights, and provide them with more continuous, comfortable, and safe riding spaces, as well as establish a good social image for riding.

Third, due to the advantages of shared e-scooters in terrain and physical fitness, the user group has expanded and changed significantly compared to shared bicycles. However, there is still much potential to adapt to users aged middle-aged and above, including improving system awareness, meeting complex lifestyle patterns, and eliminating safety concerns.

Fourth, we proposed a method to select models based on the simplicity of the model and to expand the model based on the research goal, as well as a method to screen questionnaire items to ensure that the questionnaire remains as concise as possible, thus improving questionnaire efficiency.

Finally, we acknowledge the limitations of this study in terms of data samples, random preference handling, and interpretation and prediction capabilities. We also suggest future research directions.

## Supporting information

**S1 Data. Survey results dataset.**
(XLSX)

**S1 File. Summary of scores by dimension.**
(XLSX)

**S2 File. Questionnaire template.**
(DOCX)

## Acknowledgments

We extend our heartfelt gratitude to the reviewers and editors for their invaluable comments and suggestions, which have significantly enhanced the quality of this paper. We also express our appreciation to our colleagues and friends for their unwavering support and constructive feedback throughout the research process.

## Author Contributions

**Conceptualization:** Xingjian Xue.

**Data curation:** Chenyue Lin.

**Formal analysis:** Chenyue Lin.

**Investigation:** Zhixuan Zhu.

**Methodology:** Xingjian Xue.

**Project administration:** Yue Luo.

**Resources:** Zhixuan Zhu.

**Software:** Chenyue Lin.

**Supervision:** Rui Song.

**Validation:** Yue Luo.

**Writing – review & editing:** Xingjian Xue, Rui Song.

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
