## [Decision Letter · Decision Letter 0]

1 May 2024

PONE-D-23-43995Factors related to the intention of choosing Shared E-scooters for Metro transfer: A survey study integrating weather perception into satisfaction evaluation from ChangshaPLOS ONE

Dear Dr. Xue,

Thank you for submitting your manuscript to PLOS ONE. After careful consideration, we feel that it has merit but does not fully meet PLOS ONE’s publication criteria as it currently stands. Therefore, we invite you to submit a revised version of the manuscript that addresses the points raised during the review process.

We look forward to receiving your revised manuscript.

Kind regards,

Quan Yuan, Ph.D.

Academic Editor

PLOS ONE

Journal Requirements:

The name of the colleague or the details of the professional service that edited your manuscript.A copy of your manuscript showing your changes by either highlighting them or using track changes (uploaded as a *supporting information* file).A clean copy of the edited manuscript (uploaded as the new *manuscript* file).

3. For studies reporting research involving human participants, PLOS ONE requires authors to confirm that this specific study was reviewed and approved by an institutional review board (ethics committee) before the study began. Please provide the specific name of the ethics committee/IRB that approved your study, or explain why you did not seek approval in this case.

4. Please provide additional details regarding participant consent. In the ethics statement in the Methods and online submission information, please ensure that you have specified what type you obtained (for instance, written or verbal, and if verbal, how it was documented and witnessed). If your study included minors, state whether you obtained consent from parents or guardians. If the need for consent was waived by the ethics committee, please include this information.

"This work was supported by the Natural Science Foundation of Hunan Province, China (Grant No. 2022JJ31017)."

Please note that funding information should not appear in the Acknowledgments section or other areas of your manuscript. We will only publish funding information present in the Funding Statement section of the online submission form. Please remove any funding-related text from the manuscript. 

6. We note that your Data Availability Statement is currently as follows: 

"All relevant data are within the manuscript and its Supporting Information files."

8. We note that Figure 1 in your submission contain copyrighted images. All PLOS content is published under the Creative Commons Attribution License (CC BY 4.0), which means that the manuscript, images, and Supporting Information files will be freely available online, and any third party is permitted to access, download, copy, distribute, and use these materials in any way, even commercially, with proper attribution. For more information, see our copyright guidelines: http://journals.plos.org/plosone/s/licenses-and-copyright.

1) You may seek permission from the original copyright holder of Figure 1 to publish the content specifically under the CC BY 4.0 license. 

2) If you are unable to obtain permission from the original copyright holder to publish these figures under the CC BY 4.0 license or if the copyright holder’s requirements are incompatible with the CC BY 4.0 license, please either i) remove the figure or ii) supply a replacement figure that complies with the CC BY 4.0 license. Please check copyright information on all replacement figures and update the figure caption with source information. 

If applicable, please specify in the figure caption text when a figure is similar but not identical to the original image and is therefore for illustrative purposes only.

**Additional Editor Comments:**

Please try to improve your paper and respond to all the reviewers' comments.

Reviewers' comments:

Reviewer's Responses to Questions

**Comments to the Author**

1. Is the manuscript technically sound, and do the data support the conclusions?

Reviewer #1: Yes

Reviewer #2: Yes

2. Has the statistical analysis been performed appropriately and rigorously? 

Reviewer #1: Yes

Reviewer #2: Yes

3. Have the authors made all data underlying the findings in their manuscript fully available?

Reviewer #1: Yes

Reviewer #2: Yes

4. Is the manuscript presented in an intelligible fashion and written in standard English?

Reviewer #1: Yes

Reviewer #2: Yes

5. Review Comments to the Author

Reviewer #1: The manuscript sounds technically and uses questionnaire data for statistical analysis, which can basically support the conclusions. The authors have provided a detailed description of the data processing, and presented it in the form of simple tables. However，some issues should be considered. 1. the writing of the manuscript needs improvement with the assistance of English editing, cause some grammatical, styling, and typos are found in the manuscript. 2. the format is inconsistent when arranging the introduction, eg. the interrogative sentence in the second paragraph of (1) should be placed at the end of (1) to enhance the readability of the manuscript. 3. the reference format needs to be optimized (e.g. reference [46] is incomplete), and the clarity of the images needs to be improved through the manuscript; 4. it is necessary to point out the link source when citing data; 5. the suggestion to set up a changing room should be explained more detailed, which needs to be specifically considered; 6. at the beginning of the manuscript, gender, age, occupation, and other characteristics are distinguished in the sample features, but the differences between different feature samples are not considered in the subsequent analysis; 7. how to test the applicability and effectiveness of the M4 model among different samples should be detailed.

Reviewer #2: 1.The quality of the current images needs improvement. High-quality images can better showcase the content.

2.The table analysis in the article's conclusion is not very intuitive and could be presented more clearly using more figures and graphics.

3.The utilized data only spans two days. Is this sufficient to this study?

4. The article mentions that the sample includes a high proportion of young people and participants with higher levels of education, which may affect the representativeness of the results. It is recommended that the authors discuss the potential impact of sample bias on the research findings.

5.The current method is the Technology Acceptance Model (TAM). Maybe other analytical methods can be added for comparison to reflect the universality of the analysis?

6.Could the authors discuss potential limitations of applying the TAM model to the context of Shared E-scooter (SE) usage?

7.There are also some minor issues:

-Improve the overall layout quality of the document, such as ensuring that Tables 2-6 do not span across pages.

-Table 9 is too wide.

6. PLOS authors have the option to publish the peer review history of their article (what does this mean?). If published, this will include your full peer review and any attached files.

Reviewer #1: No

Reviewer #2: No

---

## [Author Response · Author response to Decision Letter 0]

30 May 2024

1.We have ensured that the manuscript meets PLOS ONE's style requirements.

2.We have made improvements to the manuscript and had an English editor revise our article. All changes are marked and do not affect the content or structure of the paper. We sincerely appreciate the efforts of the Editors and Reviewers and hope the revisions meet with your approval.

3.We have detailed the content of the participant's consent in the manuscript and in the "Ethics Statement" field.

4.We have detailed the content of the participant's consent in the manuscript and in the "Ethics Statement" field.

5.We have removed Funding-related text from the Acknowledgements section.

6.We confirm that the entirety of the" minimal data set" has been covered in the latest submission of manuscript and "Supporting Information".

7.We have detailed the content of the participant's consent in the manuscript and in the "Ethics Statement" field.

8.Copyrighted images have been removed from the latest submission of the manuscript.

Thank you for your time and effort!

---

## [Decision Letter · Decision Letter 1]

22 Aug 2024

Factors related to the intention of choosing Shared E-scooters for Metro transfer: A survey study integrating weather perception into satisfaction evaluation from Changsha

PONE-D-23-43995R1

Dear Dr. Xue,

We’re pleased to inform you that your manuscript has been judged scientifically suitable for publication and will be formally accepted for publication once it meets all outstanding technical requirements.

Kind regards,

Quan Yuan, Ph.D.

Academic Editor

PLOS ONE

Additional Editor Comments (optional):

Reviewers' comments:

Reviewer's Responses to Questions

**Comments to the Author**

1. If the authors have adequately addressed your comments raised in a previous round of review and you feel that this manuscript is now acceptable for publication, you may indicate that here to bypass the “Comments to the Author” section, enter your conflict of interest statement in the “Confidential to Editor” section, and submit your "Accept" recommendation.

Reviewer #1: All comments have been addressed

Reviewer #2: All comments have been addressed

2. Is the manuscript technically sound, and do the data support the conclusions?

Reviewer #1: Yes

Reviewer #2: Yes

3. Has the statistical analysis been performed appropriately and rigorously? 

Reviewer #1: Yes

Reviewer #2: Yes

4. Have the authors made all data underlying the findings in their manuscript fully available?

Reviewer #1: Yes

Reviewer #2: Yes

5. Is the manuscript presented in an intelligible fashion and written in standard English?

Reviewer #1: Yes

Reviewer #2: Yes

6. Review Comments to the Author

Reviewer #1: The authors have carefully completed the revisions，I agree to accept the paper。However, please refer to the format requirements of the journal and determine whether the title of the chart or figure should be placed in the center.

Reviewer #2: No further questions. Good revision. The author has addressed each question comprehensively and effectively.

7. PLOS authors have the option to publish the peer review history of their article (what does this mean?). If published, this will include your full peer review and any attached files.

Reviewer #1: No

Reviewer #2: No

---

## [Editor Report · Acceptance letter]

29 Aug 2024

PONE-D-23-43995R1 

PLOS ONE

Dear Dr. Xue, 

I'm pleased to inform you that your manuscript has been deemed suitable for publication in PLOS ONE. Congratulations! Your manuscript is now being handed over to our production team.

Kind regards, 

on behalf of

Dr. Quan Yuan 

Academic Editor

PLOS ONE